# CPACK: An Intelligent Cyber-Physical Access Control Kit for Protecting Network [note 1]

**DOI:** 10.3390/s22208014

**Published:** 2022-10-20

**Authors:** Haisheng Yu, Zhixian Liu, Sai Zou, Wenyong Wang

**Affiliations:** 1University of Electronic Science and Technology of China, 2006 Xiyuan Avenue, Chengdu 611731, China; 2Macau University of Science and Technology, Wei Long Road, Taipa 999078, Macau; 3Guizhou University, College of Big Data and Information Engineering, Huaxi Load, Guiyang 550025, China

**Keywords:** Access Control List (ACL), Software-Defined Networking (SDN), security, floodlight, ONOS

## Abstract

Access Control Lists (ACL) are critical to protecting network and cyber-physical systems. Traditional firewalls mostly use reactive methods to enforce ACLs, so that new ACL updates cannot take effect immediately. In this paper, based on our previous work, we propose CPACK, an intelligent cyber-physical access control kit, which uses a smart algorithm to upgrade the ACL list. CPACK adopts a proactive way to enforce ACL and reacts to a new ACL update and network view update in real time. We implement CPACK on both Floodlight and ONOS controller. We then conduct a large number of experiments to compare CPACK with the Floodlight firewall application. The experimental results show that CPACK has a better performance than the existing Floodlight firewall application. CPACK is also integrated into the new version of Floodlight and ONOS controller.

## 1. Introduction

The Internet, accommodating a variety of heterogeneous networks and distributed applications [1], has achieved great success and has been an enormous power in promoting social and economic development since it was proposed [2]. However, the current Internet environment has changed dramatically as a result of the emerging network services and cyber phsical system scale expansion. The traditional architecture of Internet has exposed serious deficiencies, such as unexpected delays for data communication [3] and difficulty in the traffic load balance among links [4]. The fundamental reason for that is the tight coupling of control logic and data forwarding in network devices (e.g., router, switch) and the distributed control of network devices [5].

To solve the problem, OpenFlow [6] has been proposed, which strips the control logic from network devices and provides a set of well-defined interfaces for programming. OpenFlow facilitates the realization of new network protocols and topologies without changing network devices. The concept of Software-defined Networking (SDN) [7] is created on the basis of OpenFlow and other research work such as SANE [8] and Ethane [9]. SDN migrates the control logic in network devices to individual computing devices called controller to enforce centralized control. The control plane and forwarding plane in SDN is connected using a standard interface protocol (e.g., OpenFlow). SDN provides an open software programmable model and a diversity of network control functions [10]. It has gained wide recognition and good support from both academia and industry.

Access Control List (ACL) is a network security enhancement. It applies a set of ACL rules to each IP packet and determines whether to forward or drop the packet based on its header fields [11]. ACL is similar to the stateless firewall or packet filtering firewall which provides basic traffic filtering capabilities [12]. In traditional networks, ACL is often placed in network devices (e.g., router, switch) and can be configured to control both inbound and outbound traffic. Network devices examine each packet and determine whether to forward or drop the packet on the basis of the rules specified in ACL [13]. Unfortunately, the approach has several deficiencies. Firstly, network devices should have appropriate hardware and processing capabilities to enforce ACL, causing a vast expense. Moreover, it is too complicated to design and configure ACL in distributed network devices, not to mention the situation when network security policy changes. The cumbersome maintenance of ACL in complex networks is also prone to error.

The root reason for this issue lies in the distributed way of enforcing ACL in traditional networks. Software-defined Networking (SDN) provides a convenient network paradigm to solve the problem [14]. SDN separates control logic and forwarding logic in traditional networks, and SDN controller configures networks in a centralized manner rather than distributed configuration [15].

There have been a few studies focusing on ACL in SDN up to now. D. Gamayunov et al. proposed an approach to migrate ACL in traditional networks to SDN with security policy preservation [16]. However, the approach chooses SDN topology based on how the subnets are divided in traditional networks; therefore, it cannot adapt to the situation that a specified SDN topology is given. Shin et al. designed an OpenFlow security application development framework for developing OpenFlow-enabled detection and mitigation applications by modular composition [17]. It aims at a holistic platform for implementing new security applications which is much more sophisticated than ACL. Trandafir et al. presented FirewallPK, a centralized Access Control List management application [18]. The application was developed on the Cisco One Platform Kit framework and requires specified Cisco network devices. X. Jia et al. proposed SDN based distributed firewalls for P2P networks, while the firewalls do not leverage the centralized control feature which SDN provides [19]. Francois et al. reviewed approaches which implement in-network security functions such as firewalls supported through OpenFlow devices [20]. Generally, two ways are adopted to enforce ACL in SDN: reactive and proactive.

In a reactive way, controller compares each incoming Packet-in message against all ACL rules from the highest priority until either a match is found or the list is exhausted. When a match is found and the ACL rule denies the flow, the controller pushes a drop flow entry; otherwise it pushes a regular forwarding flow entry [21]. Unfortunately, this way shows defects in several aspects. Firstly, as a controller starts a comparison process for each new Packet-in message, it increases forwarding delay. Secondly, there is frequent interaction between switches and controller, which takes up a large portion of the controller’s resources, especially when network traffic bursts. Moreover, the worst issue is caused when a new ACL update occurs, as the reactive way cannot remove invalid flow entries in time, resulting in a new ACL update that cannot take effect immediately, such that the delay is unpredictable. For example, in Figure 1, suppose that Alice is sending packets to Bob, and there is a flow entry forwarding all packets from Alice to Bob. When a new added ACL rule denies all packets from Alice to Bob, that flow entry becomes invalid immediately but is not removed. Therefore, Alice is still able to send packets to Bob as long as the invalid flow entry remains in the switch. That is to say, the reactive way can cause network security violation. M. Suh, J. Collings, T Javidet et al. implemented ACL applications in a reactive way [22,23,24]. Timothy L. Hinrichs et al. proposed a declarative policy language FML [25], which allows users to declare policy such as ACL rules. The language is reactive and needs the policy engine to intercept every flow on the network. The famous Floodlight [26] controller contains a firewall application [27] working in a reactive way, too.

In a proactive way, the controller uses flow entry to block packets in switch in advance, without being requested by Packet-in messages [28]. The proactive way avoids additional forwarding delay and saves the controller’s resources. Justin Gregory V. Pena et al. proposed a distributed flow-based firewall working in a proactive way [29]. The application listens to the switch’s ConnectionUp events and pushes flow entry for ACL rules when a new switch appears in the network. In the application, new added ACL rules do not take effect in the switches that have already shown up, because no ConnectionUp events are raised for those switches.

In our earlier work [30], we proposed a user-driven centralized ACL framework in SDN, which adopts a proactive way to enforce ACL and thus to avoid additional delay and save the controller’s resource. It reacts to a new ACL update and network view update in real time to ensure network security. In this paper, we propose CPACK, an intelligent Cyber-Physical Access Control Kit, which use smart algorithm to upgrade the ACL list. CPACK uses an abstract network view to accelerate processing and undertakes a match check for new added ACL rule to avoid the invalid rule. We elaborate on the update process and upgrade algorithm of ACL in CPACK, and also describe the mandatory ACL update, AP pair update, and the order of adding ACL rules to ensure that CPACK meets the needs of users.We implement CPACK on both Floodlight and ONOS controller [31], and CPACK is also integrated into the new version of both controllers.

## 2. CPACK DESIGN

### 2.1. Overview

Figure 2 depicts CPACK’s architecture, CPACK provides REST API for users and contains two core modules, Accessing Pair (AP) Manager and Access Control List (ACL) Manager. Each module has several submodules in charge of different processing.

In CPACK, each ACL rule contains several match fields and an action field. Packets defined in match fields are forwarded or dropped following the action field. An ACL rule is denoted as:


*R: {id; nw proto; src ip; dst ip; dst port; action}*


Each ACL rule has a distinct id. Match fields comprises *nw_proto* (network protocol), *src_ip* (source IP address), *dst_ip* (destination IP address), *dst_port* (TCP or UDP destination port). Match field value may be a wildcard, which can be substituted for all possible field values. The *src_ip* and *dst_ip* field use CIDR IP address, which can designate many unique IP addresses. The *action* field value is either “ALLOW” or “DENY”.

CPACK provides a friendly and centralized user interface through REST API for users to add, remove, and query ACL rules. Users can use CPACK easily by sending an HTTP request containing JSON string, and they do not need to configure distributed switches one by one any more because CPACK does all the work.

CPACK filters IP packets by ACL flow entries exactly reflecting ACL rules in ingress or egress switches. After receiving the user’s new ACL update request, CPACK updates ACL rules and ACL flow entries immediately.

We will describe CPACK’s core modules in the following subsections.

### 2.2. Accessing Pair Manager

In CPACK, the real network view is transformed to an abstract network view. The abstract network view conceals internal network topology, and it only exposes the interfaces between edge switches and external hosts in the networks, as Figure 3 depicts.

We use Accessing Pair (AP) to store the interface information in the abstract network view. An AP is denoted as:

AP: {id,dpid,ip}

The fields represent AP id, edge switch’s dpid (data path id), and host’s IP address, respectively.

Accessing Pair Manager is a CPACK module, which maintains AP information in real time and provides a query function. AP Manager monitors host an update event in the networks and store all interface information in AP Set.

When a new host appears or disappears in the networks, AP Manager updates AP Set correspondingly and calls ACL Manager for further processing, which will be described in Section 2.3.

AP Manager also provides a query function *getSwitchSet*. Given a CIDR IP address, the function traverses AP Set and returns a switch set. Each switch in the set connects with a host whose IP address is contained in the CIDR IP address. This function will be used when generating ACL flow entry.

### 2.3. Access Control List Manager

Access Control List (ACL) Manager is a CPACK module, which updates ACL, enforces the update, and processes the AP update.

#### 2.3.1. Intelligent Update ACL

In order to make ACL update intelligently, we use the following intelligent update algorithm of ACL, as described in Algorithm Section 2.3.1.

After receiving a new ACL update request, ACL Manager verifies its validity and returns an error message if not valid.

If a user requests to add a new ACL rule, ACL Manager firstly parses user’s request JSON string and generates a new ACL rule. It then traverses ACL Rule Set to check whether the new ACL rule matches another existing rule. The new rule is rejected if a match is found. ACL Manager generates a distinct id for each rule passing match check, adds it to ACL Rule Set and starts the enforcing stage.
**Algorithm 1** Updating ACL
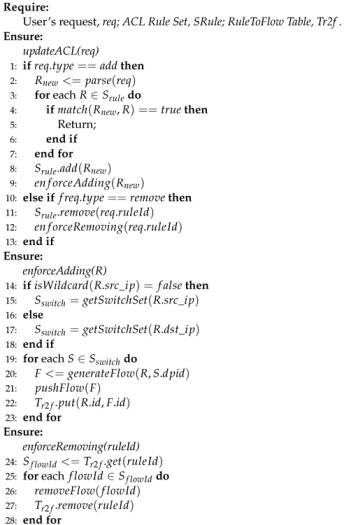



Match check is important because it rejects invalid rules in order to reduce storage overhead in both switches and controller. Two functions are used in match check, and they give the definition of *match*:

*cover($R_{new}$,$R_{old}$,field)*: A Boolean function, where $R_{new}$, $R_{old}$ denote ACL rules and *field* denotes ACL rule’s match field. We say:

*cover(R_new_,R_old_,field) = true* if:

For fieldϵ{nw_proto,dst_portg}, $R_{old}$.*field* has a wildcard value, and $R_{new}$.*field* has an user-assigned value;

For fieldϵ{src_ip;dst_ip}, $R_{old}$.*field* contains all the IP addresses in $R_{new}$.*field*.

match($R_{new}$,$R_{old}$): A Boolean function. We say:

*match(R_new_;R_old_) = true* if:

For fieldϵ{nw_proto,src_ip,dst_ip,dst_port}, there is:

$R_{new}$.*field* = $R_{old}$.*field* or *cover*($R_{new}$,$R_{old}$.*field*) *= true*.

We say ACL rule $R_{new}$ matches $R_{old}$ if all packets filtered by $R_{new}$ is already filtered by $R_{old}$, and $R_{new}$ will not work at all if added.

If the user requests to remove an existing ACL rule, ACL Manager firstly parses the user’s request and gets the rule’s id. It then removes the rule from ACL Rule Set and starts the enforcing stage.

#### 2.3.2. Enforce ACL Update

After updating ACL, the ACL Manager enforces that update by ACL flow entries. An ACL flow entry is a static flow entry generated by CPACK to enforce ACL, denoted as:


*F: {id, priority, dpid, nw_proto; src_ip, dst_ip, dst_port, action}*


After a new ACL rule is added, CPACK enforces that rule. Generally, CPACK pushes ACL flow entries in ingress switches where packets enter the networks. However, if an ACL rule’s *src_ip* field has a wildcard value, it is reasonable to push ACL flow entries into egress switches where packets leave the networks. Therefore, the ACL Manager should pass different IP addresses to function *getSwitchSet* provided by AP Manager module, depending on whether the rule’s *src_ip* field has a wildcard value or not. The function returns a set containing all the switches where CPACK should push ACL flow entries into.

For each switch *S* in the result set, the ACL Manager generates an ACL flow entry F enforcing the new added rule *R*. The flow entry’s priority is carefully set to ensure earlier generated ACL flow entry in switches always has higher priority. Other fields are:•F.dpid⟸S.dpid•F.field⟸R.field, fieldϵ{nw_proto,src_ip,dst_ip,dst_port}•F.action⟸ “DROP” **if** *R.action* = “DENY”•F.action⟸ “FORWARD to CONTROLLER” **if** *R.action* = “ALLOW”

ACL Manager pushes every new generated ACL flow entry into the right switch and stores the mapping from ACL rule to ACL flow entry in a hash table called *RuleToFlow (r2f)* Table. Then, the new added rule starts to take effect.

By ACL flow entries exactly reflecting new added ACL rule in appropriate ingress or egress switches, CPACK enforces the ACL rule correctly and efficiently.

After an existing ACL rule is removed, the ACL Manager searches *RuleToFlow* Table and gets ACL flow entries relevant to that rule. It then removes those flow entries from switches and updates *RuleToFlow* Table.

#### 2.3.3. Process AP Update

As is mentioned before, when a new AP update occurs in the networks, AP Manager module calls ACL Manager module for further processing.

If a new host appears, ACL Manager traverses ACL Rule Set to check whether the host’s IP address is contained in an ACL rule’s *src_ip* and *dst_ip* field. As we want to make sure whether ACL flow entry should be pushed into ingress switches or egress switches, it is necessary to firstly check whether the ACL rule’s *src_ip* field has a wildcard value. If the ACL rule has a wildcard value in *src_ip* field, ACL Manager compares the host’s IP address with *dst_ip* field, or *src_ip* field otherwise. If a relevant ACL rule is found, ACL Manager generate a new ACL flow entry and pushes it to the switch connecting with the new host.

If an existing host disappears, ACL Manager removes ACL flow entry relevant to that host. This is easy because the ACL Manager maintains a hash table storing the mapping from AP to ACL flow entry.

#### 2.3.4. Order to Add ACL Rules

As CPACK enforces new ACL rules instantly when they are added, earlier added ACL rule always has higher priority. CPACK sets ACL flow entry’s priority carefully to ensure the priority definition in ACL rule. Therefore, the order to add ACL rules is critical.

CPACK allows all IP packets by default, and an ACL rule is generally used to deny packets. Note that if an ACL rule’s *action* field value is “ALLOW”, the rule is used to forward a subset of packets dropped by another rule whose *action* field is “DENY”. Figure 4 depicts an example to show the vital importance of the order to add allowing ACL rules.

Suppose there are three hosts and one web server in the network. The networking administrator intends to allow Alice but deny Bob and Chuck to access the web server on port 80.

In (a), the administrator firstly adds an ACL rule *Rule1* to allow Alice’s access; accordingly, CPACK pushes an ACL flow entry *Flow1* in switch S1 where Alice accesses the network. The administrator then adds *Rule2* to deny Bob and Chuck, and CPACK pushes *Flow2* and *Flow3* in *S2* and *S3*. Note that in *S1*, *Flow1* has higher priority than *Flow2*. When Alice requests to access the web server, *Flow1* forwards Alice’s first packet to controller in a Packet-in message. The controller then pushes general forwarding flow entries for Alice. To ensure validity, the forwarding flow entries must have larger priority than all ACL flow entries. When Bob and Chuck request to access the web server, *Flow2* and *Flow3* drop their requests directly in switches.

However, if the administrator adds the two ACL rules in an inverted order, like in (b), *Rule2* can not pass match check because it matches *Rule1*. As a result, CPACK rejects *Rule1*, and all hosts in subnet 10.0.0.0/30 (including Alice) are unable to access the web server, which violates the administrator’s intention.

## 3. Implementation and Evaluation

CPACK is a logically centralized ACL, while it can be physically distributed. We implement CPACK in two versions: a single-controller version on Floodlight controller and multi-controller version on ONOS controller. Both are integrated into the latest version of Floodlight and ONOS controller.

The initial implementation of CPACK on Floodlight controller is based on a single controller for simplicity. However, when the number and size of production networks deploying OpenFlow increases, a single controller for the entire network will expose several defects such as large amount of control traffic, long flow setup times and latencies [32]. Therefore, we extend CPACK to a multi-controller version on ONOS controller. Compared to the single-controller version, the multi-controller version CPACK provides scalability while guaranteeing that the ACL is logically centralized, and it is also resilient to controller failures.

The single-controller version CPACK is implemented as a Floodlight module using Java. CPACK uses Floodlight’s *IRestApiService* to provide REST API and uses Floodlight’s *IStorageSourceService* to manage the flow entry in switches. CPACK adds itself as a listener to Floodlight’s *IDeviceService* service and will be notified if a new host event appears. CPACK also utilizes several methods provided by Floodlight to operate on the CIDR IP address.

The multi-controller version CPACK is implemented as an ONOS controller application using Java. The CPACK application runs on each controller in an ONOS cluster, and the ACL data is synchronized among all the CPACK applications using Raft consensus algorithm. Therefore, all the CPACK applications share the same consistent ACL data and can serve user’s ACL update request locally. When a user requests a new ACL update in any controller in an ONOS cluster, the CPACK application running on that controller would map the new ACL update to the ACL flow entry update and propagate both the updates among the controller cluster. The right controller would then finish the ACL flow entry update in the switch. Since the ACL data in all CPACK applications is consistent, when one of the controllers in the cluster fails, another controller can take over it without losing any ACL data.

The multi-controller version CPACK stores data using ONOS’s *StorageService*, all the data is stored in one of the ONOS distributed primitives, *ConsistentMap*, which guarantees strong consistency and ensures good scale out characteristic [33]. CPACK listens to ONOS’s *HostService* for new host event and manages flow entry using ONOS’s *FlowRuleService*.

We compare CPACK with the Floodlight firewall application. As is mentioned before, to enforce ACL, CPACK works in a proactive way while Floodlight adopts a reactive way. This means that different events trigger their ACL enforcing process and the user’s request for CPACK and Packet-in message for Floodlight firewall application; therefore, it is unreasonable to compare their performance in general situation. We create a situation such that a new ACL update conflicts with ACL flow entry in switches, and compare the delay for a new ACL update to take effect, like in Figure 1.

We build a virtual network in Mininet [34] and run several experiments. For each experiment, we add different numbers of ACL rules in advance and insure that CPACK has to traverse ACL Rule Set during an update. Then we let host A in the network send ICMP packets to host B using *Ping* command. If host A succeed to *Ping* host B at first, we add a new ACL rule to deny the flow and record the delay until an ACL flow entry drops the flow. If there is already a ACL rule denying the flow and host A fails to *Ping* host B at first, we then remove that ACL rule and record the delay until an regular flow entry forwards the flow. The experimental result is shown in Figure 5.

The delay in the Floodlight firewall application is more than 5000 ms because a flow entry’s default idle timeout is set to 5000 ms in Floodlight, and no Packet-in messages is sent to the controller as long as the ACL flow entry persists. As a result, a new ACL update will not take effect at all until after at least an idle timeout. We regard the delay as 5000 ms uniformly in Figure 5.

As Figure 5 shows, in the single-controller version, the delay for rule adding and removing in CPACK goes up linearly as the existing ACL rule number increases because CPACK needs to traverse ACL Rule Set. The delay for enforcing ACL update vibrates for reason that CPACK needs to communicate with switches, and the delay depends on the network quality at that time.

The evaluation result for the multi-controller version is mostly similar to the single-controller version, except that the delay for rule removing remains almost unchanged. This is because we use hash tables rather than a single set to store ACL rules, and hash tables are move effective when processing indexing and updating.

The comparison result indicates that CPACK is considerably better than the Floodlight firewall application when handling new ACL update requests at the collision situation.

Figure 6 illustrates the performance of different processes in the two CPACK versions. The figure shows that the delay for all processes (except for rule removing and enforcing process in the multi-controller version) increase almost linearly but slowly as the existing ACL rule number increases sharply. It is obvious that CPACK causes little delay, and its performance is pleasant. For the same process, the multi-controller version spends more time than the single-controller version because it uses ONOS’s distributed store service, and the operations on the distributed store takes more time than that on the local store. Therefore, the single-controller version CPACK has a better performance in the scenario where only one controller is deployed, while in a multiple controller deploying scenario, the multi-controller CPACK is a better choice for high availability and good scale-out characteristic.

## 4. Conclusions and Future Work

In this paper, we propose CPACK, an intelligent Cyber-Physical Access Control Kit, which uses a smart algorithm to upgrade the ACL list. CPACK adopts a proactive method and intelligent algorithm to enforce ACL and reacts to new ACL updates and network view updates in real time. CPACK can avoid additional delay, save controller resources, and also ensure network security. We implement CPACK on Floodlight and ONOS controller. We then conduct a large number of experiments to compare CPACK with the Floodlight firewall application. The experimental results show that CPACK has a better performance than the existing Floodlight firewall application. P. Porras et al. proposed the dynamic flow tunneling scenario, which clearly shows that malicious application can evade ACL by simply adding a few flow entries in SDN [35]. The main reason for this is that OpenFlow allows various Set-Field actions that can dynamically change the packet headers [36]. P. Kazemian proposed a real time policy checking tool called *NetPlumber* [37] based on Header Space Analysis [38]. We intend to add security check capability based on HSA in CPACK to prevent attacks from adversaries in the future.

## Figures and Tables

**Figure 1 sensors-22-08014-f001:**
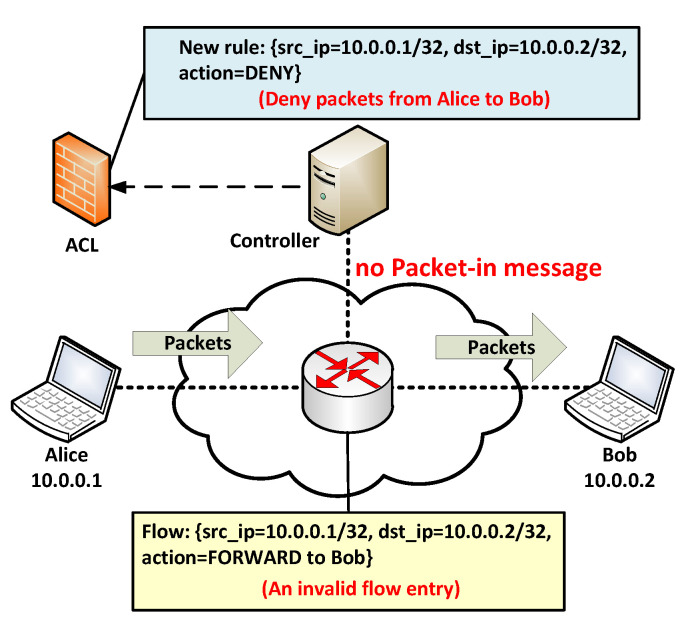
Network security violation in a reactive way.

**Figure 2 sensors-22-08014-f002:**
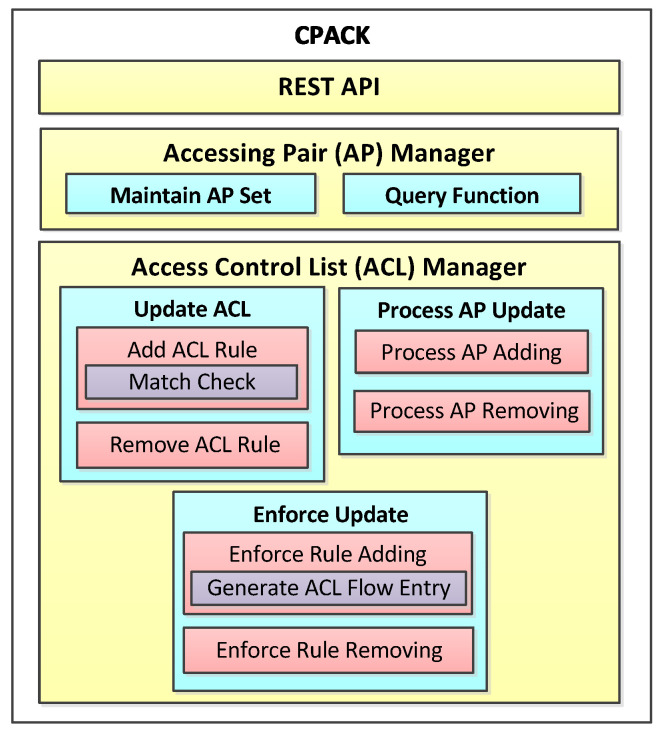
CPACK architecture.

**Figure 3 sensors-22-08014-f003:**
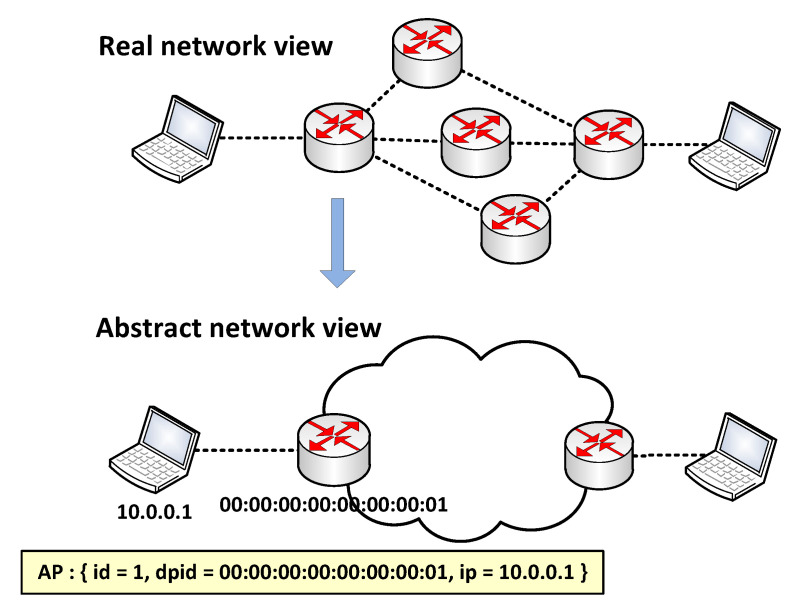
Abstract network view and Accessing Pair (AP).

**Figure 4 sensors-22-08014-f004:**
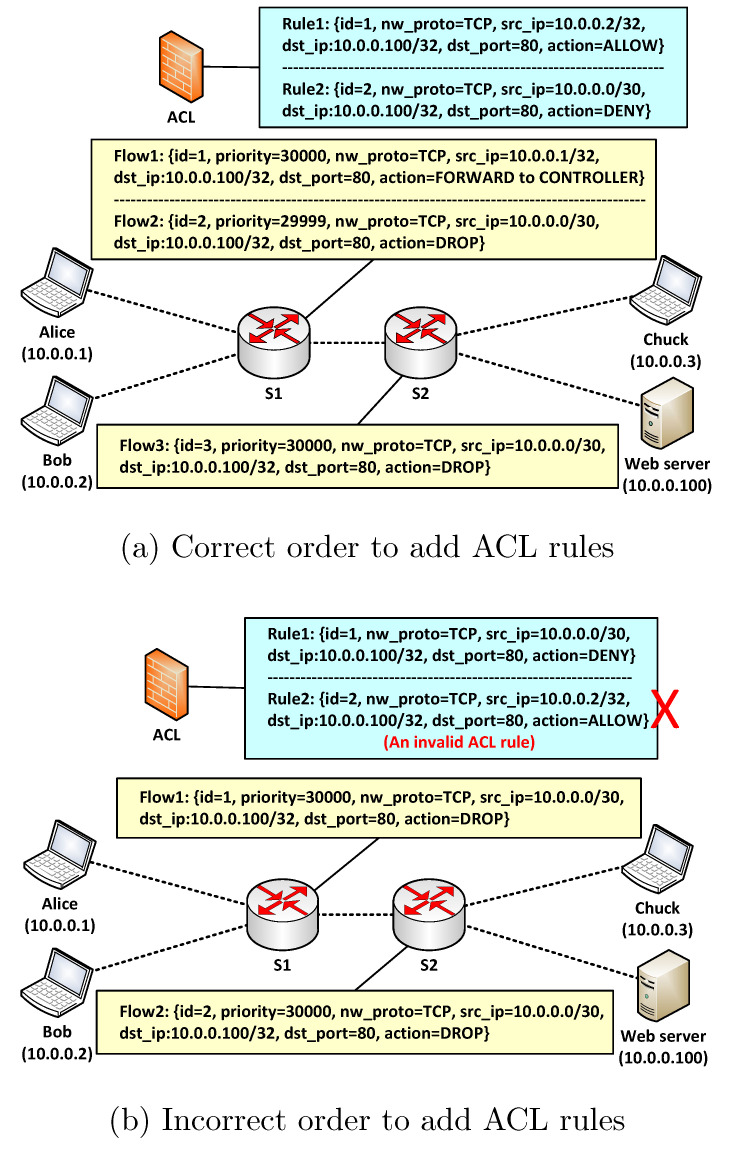
An example for the order to add ACL rules.

**Figure 5 sensors-22-08014-f005:**
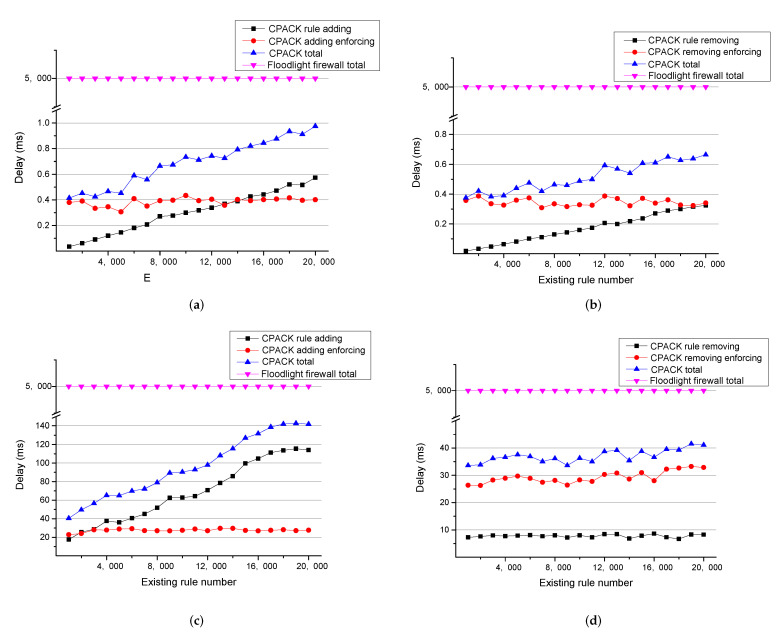
ACL update delay comparison. (**a**) Add a new ACL rule (single-controller version); (**b**) Remove an existing ACL rule (single-controller version); (**c**) Add a new ACL rule (multi-controller version); (**d**) Remove an existing ACL rule (multi-controller version).

**Figure 6 sensors-22-08014-f006:**
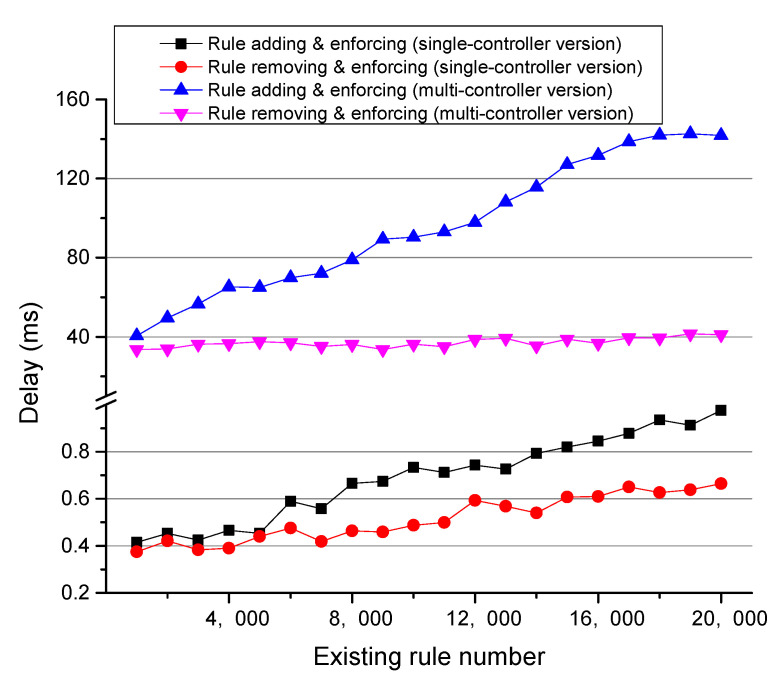
CPACK performance.

## Data Availability

Not applicable.

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
