# Peer review of "CPACK: An Intelligent Cyber-Physical Access Control Kit for Protecting Network"

_sensors, 2022, doi:10.3390/s22208014_

Round 1

Reviewer 1 Report

This paper investigates access control lists in networks, and proposes an intelligent cyber-physical access control kit. Several main comments are given as follows.

1. In Introduction, the main motivation and contribution of this paper are not clear. What is the solved problem of this paper?

2. In Section 2.3.1, how to show intelligence?

3. Why to name Cyber-physical Access Control Kit as CLACK rather than CPACK?

4. The proposed method is a proactive way, while the compared method seems a reactive way.

5. To actively deal with unexpected delays, typical control methods, for example, 10.1007/s11424-021-0160-y, 10.1109/TCSII.2021.3106694, 10.1002/rnc.5106, and 10.1016/j.amc.2021.126844 are suggested to be reviewed.

6. The English presentation of this paper should be thoroughly polished. And some typos should be corrected, for example, “which use smart algorithm” in Abstract should be “which uses smart algorithm”, “The fundamental reason for that” should be “The fundamental reason for those”, and so on.

Author Response

We are extremely grateful for the We are extremely grateful for the valuable suggestions and comments from the reviewers. We have carefully addressed each comment in the reviews. Below we summarize our revision based on these reviews.

Reviewer 1

This paper investigates access control lists in networks, and proposes an intelligent cyber-physical access control kit. Several main comments are given as follows.

Re:Thanks for your comments.

  • In Introduction, the main motivation and contribution of this paper are not clear. What is the solved problem of this paper?

Re: As we mentioned in the introduction, this paper introduces the CPACK, a cyber-physical access control kit, which adopts a proactive way to enforce ACL thus avoiding additional delay and saving the controller’s resources.

  • In Section 2.3.1, how to show intelligence?

Re: Our ACL update algorithm can add and remove ACLs according to user requirements. The matching algorithm and the regulation of writing order prevent invalid rules from being written, which not only ensures the rationality of ACL but also saves the storage loss of the controller and switch.

  • Why to name Cyber-physical Access Control Kit as CLACK rather than CPACK?

Re:We are glad to accept your suggestion.

  • The proposed method is a proactive way, while the compared method seems a reactive way.

Re:We have listed both reactive way and proactive way of ACL update methods to ensure that you are aware of the various methods used in the industry.

  • To actively deal with unexpected delays, typical control methods, for example, 10.1007/s11424-021-0160-y, 10.1109/TCSII.2021.3106694, 10.1002/rnc.5106, and 10.1016/j.amc.2021.126844 are suggested to be reviewed.

Re:Thanks for your suggestion.

  • The English presentation of this paper should be thoroughly polished. And some typos should be corrected, for example, “which use smart algorithm” in Abstract should be “which uses smart algorithm”, “The fundamental reason for that” should be “The fundamental reason for those”, and so on.

Re:Thank you for your suggestions, we have thoroughly checked and revised the English in the article.

Reviewer 2 Report

Presented article of authors titled "CLACK: an intelligent Cyber-physicaL Access Control Kit for protecting network" is a contribution in the field of the SDN network security and traffic filtering. The article proposes new algorithm for updating and modifying ACL lists. The solution was tested in a virtual lab scenario, on two different SDN controllers.

The article consists of five chapters. It contains 12 pages of text, including the list of 38 references, 6 figures and one algorithm.

The abstract is reasonably extensive and sufficiently explanatory. The list of references contains also older resources, which are, in my opinion, fully sufficient.

In the first chapter, authors provide relatively in-depth introduction to all the related components used in the paper. Part of this Introduction chapter is also review of relates works. Chapter 2 proposes design of CLACK algorithm and also contains pseudocode algorithm of adding and modifying ACL rule. The third chapter deals with implementation and shows results of experiments in clear graph form. The fourth chapter concludes the paper and proposes future work. The last chapter is list of fundings.

Regarding of the paper, to the best of my knowledge, I think that the topic of research is actual and proposed solution should work not only in virtual scenario.

I have one comment/question to authors, if they could explain, why they didn’t include source port into ACL definition in CLACK design. In my opinion, it is also important part of ACL and I have seen several production ACL’s using source port as important part of rules.

In my opinion, the chosen language is at good level, the paper is readable and understandable. All objectives are clearly defined, and the steps during experiments are sufficiently documented. Considering mentioned I recommend the article for publication in the MDPI Sensors journal after answering the question above.

Author Response

We are extremely grateful for the We are extremely grateful for the valuable suggestions and comments from the reviewers. We have carefully addressed each comment in the reviews. Below we summarize our revision based on these reviews.

Reviewer 2

Presented article of authors titled "CLACK: an intelligent Cyber-physicaL Access Control Kit for protecting network" is a contribution in the field of the SDN network security and traffic filtering. The article proposes new algorithm for updating and modifying ACL lists. The solution was tested in a virtual lab scenario, on two different SDN controllers.

The article consists of five chapters. It contains 12 pages of text, including the list of 38 references, 6 figures and one algorithm.

  • The abstract is reasonably extensive and sufficiently explanatory. The list of references contains also older resources, which are, in my opinion, fully sufficient.In the first chapter, authors provide relatively in-depth introduction to all the related components used in the paper. Part of this Introduction chapter is also review of relates works. Chapter 2 proposes design of CLACK algorithm and also contains pseudocode algorithm of adding and modifying ACL rule. The third chapter deals with implementation and shows results of experiments in clear graph form. The fourth chapter concludes the paper and proposes future work. The last chapter is list of fundings.

Re:Thanks for your comments.

  • Regarding of the paper, to the best of my knowledge, I think that the topic of research is actual and proposed solution should work not only in virtual scenario.

Re:Thanks for your suggestion.

  • I have one comment/question to authors, if they could explain, why they didn’t include source port into ACL definition in CLACK design. In my opinion, it is also important part of ACL and I have seen several production ACL’s using source port as important part of rules.

Re:You raise a good question. In the actual access control list, we only control the IP address, of course, if the user has the corresponding demand, we can also write the source port into the control rule. In future research, we will add more complex control rules that take the source port into account.

  • In my opinion, the chosen language is at good level, the paper is readable and understandable. All objectives are clearly defined, and the steps during experiments are sufficiently documented. Considering mentioned I recommend the article for publication in the MDPI Sensors journal after answering the question above.

Re:Faithful Thanks for your comments.